# Passive Vision Detection of Torch Pose in Swing Arc Narrow Gap Welding

**DOI:** 10.3390/s24154996

**Published:** 2024-08-02

**Authors:** Na Su, Haojin Jia, Liyu Chen, Jiayou Wang, Jie Wang, Youmin Song

**Affiliations:** 1Jiangsu Provincial Key Laboratory of Advanced Welding Technology, School of Materials Science and Engineering, Jiangsu University of Science and Technology, Zhenjiang 212003, China; jiahaojin2003@163.com (H.J.); lychen2004gen@163.com (L.C.); jywang@just.edu.cn (J.W.); 2School of Communications and Information Engineering, Nanjing University of Posts and Telecommunications, Nanjing 210003, China; 3Kunshan Huaheng Welding Co., Ltd., Kunshan 215300, China

**Keywords:** swing arc narrow gap welding, torch pose, passive visual sensing, image processing

## Abstract

To enhance the synchronous detection of the horizontal and vertical positions of the torch in swing arc narrow gap welding, a torch pose detection (TPD) method is proposed. This approach utilizes passive visual sensing to capture images of the arc on the groove sidewall, using advanced image processing methods to extract and fit the arc contour. The coordinates of the arc contour center point and the highest point are determined through the arc contour fitting line. The torch center position is calculated from the average horizontal coordinates of the arc contour centers in adjacent welding images, while the height position is determined from the vertical coordinate of the arc’s highest point. Experimental validation in both variable and constant groove welding conditions demonstrated the TPD method’s accuracy within 0.32 mm for detecting the torch center position. This method eliminates the need to construct the wire centerline, which was a requirement in previous approaches, thereby reducing the impact of wire straightness on detection accuracy. The proposed TPD method successfully achieves simultaneous detection of the torch center and height positions, laying the foundation for intelligent detection and adaptive control in swing arc narrow gap welding.

## 1. Introduction

Swing arc narrow gap gas metal arc welding (SA-NG-GMAW) is an advanced process for welding thick-walled components [1,2,3]. In this process, the arc is periodically swung to adjust thermal distribution, thereby enhancing the fusion of the narrow gap groove sidewalls. This method offers excellent arc directionality, reduces wear on the contact tip, and is suitable for all-position welding [4,5,6].

However, factors such as groove machining errors, assembly inaccuracies, welding deformation, and variations in groove types can cause the welding torch center to deviate from the groove center during welding [7,8]. This deviation can result in uneven penetration on both sidewalls of the groove [9,10,11]. Therefore, in the SA-NG-GMAW process, it is crucial to ensure that the torch remains aligned with the groove center and maintains a consistent height [12,13]. Real-time detection of the torch’s position and orientation is essential for improving welding quality, whether welding low-carbon steel or high-strength steel [14,15,16].

To achieve an accurate narrow gap welding torch pose detection, effective sensing methods are crucial [17]. Examples include vision-based sensing [18], eddy current sensing [19], and arc sensing [20]. Among these, vision-based methods have garnered widespread attention due to their noncontact and ability to provide rich information. Vision-based sensing can be further categorized into active visual sensing and passive visual sensing [21,22].

Active visual sensing methods have demonstrated significant potential in welding applications. Zeng et al. [23] proposed a vision-aided 3D path teaching method for narrow butt joint welding, combining 2D pixel coordinates and 3D point cloud data to precisely calculate the torch’s pose. Similarly, Zhang et al. [24] developed a 3D vision-driven robotic path planning approach that includes a welding torch posture planning method based on the neighborhood centroid. Additionally, Zhu et al. [25] used a single modulated laser line image to accurately calibrate the spatial position and posture parameters of the welding torch relative to the welding groove. In another study, Li et al. [26] employed a structured light-based visual method for robotic pipe welding pose optimization, adjusting the welding torch’s pose based on the extracted features. Moreover, Zhang et al. [27] developed a vision-based adaptive welding torch pose adjustment method to ensure precise and collision-free robot motions in the wire and arc additive remanufacturing of hot-forging dies. Furthermore, Fang et al. [28] proposed a novel model-based welding trajectory planning method for identical structural workpieces using point cloud data. However, active visual sensing has drawbacks, such as premature detection and the need for additional light sources.

Compared to active visual sensing, passive visual sensing exhibits unique advantages in certain applications. Wang et al. [29] developed a passive visual plasma robotic welding system using a digital CCD camera and image processing to automatically track the weld line and maintain a consistent standoff distance, ensuring precise torch guidance and stable welding conditions. To reduce welding interferences, Zhu et al. [30] proposed an image processing algorithm to obtain the torch center position by extracting the wire edge from welding images. To improve detection accuracy, Su et al. [31] developed an optimal estimation-based position consistency filtering algorithm to detect the torch center from wire edges. To further control the welding torch’s pose, Shi et al. [32] introduced a cooperative sensing and control strategy to achieve process stability and deposition height control in robotic pulsed gas tungsten arc additive manufacturing on uneven substrates, thereby ensuring consistent welding quality. Additionally, Xia et al. [33] proposed a method for detecting the horizontal position of the welding torch using the wire center in narrow gap gas metal arc welding robots. Passive visual sensing detection approaches can synchronize detection with the arc position without an additional light source. Moreover, they can acquire rich visual information, including the weld pool, arc, wire, and groove edges, making them highly advantageous in complex welding scenarios.

To address the issues of poor synchronization and algorithmic complexity in existing methods for detecting the welding torch pose during narrow gap welding, this work proposes a new torch pose detection (TPD) method based on passive sensing. This method detects the torch pose from arc points through image processing, including a morphological anti-interference extraction method for the arc profile and detection algorithms for the torch center and height position. Additionally, actual welding experiments were conducted to demonstrate the anti-interference capability and effectiveness of the proposed approach.

This paper is organized as follows: Section 2 presents the passive visual sensing detection system of the torch pose. Section 3 describes the method for torch pose detection. Section 4 shows the experimental results of torch pose detection. Finally, Section 5 provides the conclusions. The primary purpose of this study was to enhance synchronization, real-time performance, and efficiency in torch pose detection, ultimately enhancing weld quality through real-time torch pose detection and enabling precise adjustments based on welding parameters.

## 2. Passive Visual Sensing Detection System of Torch Pose

### 2.1. System Construction

Figure 1 shows the passive visual sensing detection system for the torch position. This system comprises a swing arc narrow gap welding module, a visual sensing module, and a computer image processing module. Additionally, it includes components such as a mixed gas, a wire feeding mechanism, a current sensor, and a moving mechanism with a worktable.

The swing arc narrow gap welding module mainly consists of a swing arc narrow gap welding torch, a welding power supply, and a welding controller. Electric power is supplied to the arc via a carbon brush connected to the positive terminal of the welding power supply. The welding controller sets the arc swing angle, swing frequency, and sidewall stay time, controlling the periodic rotation of the bending conductive rod at a certain frequency. This causes the arc at the end of the wire to swing periodically within the narrow gap groove.

The visual sensing module includes a camera and a dimming system. An 8-bit camera with a frame rate of 108 fps and a resolution of 1312 × 1082 is used. The camera is fixed on the swing arc narrow gap welding torch, maintaining a constant distance between the camera lens and the end of the wire. It is aimed at the molten pool in front of the torch at a certain depression angle. The welding testpiece is fixed on the worktable of the traveling mechanism, moving forward relative to the torch at a welding speed of *V_w_*.

The computer image processing module consists of a PC, an image acquisition card, and image processing algorithms. The image acquisition card transmits the captured welding images to the PC. The PC then uses image processing algorithms to extract the fitting line of the arc contour and calculate the left, right, upper (highest point of the arc), and lower limit points of the arc contour. This enables the detection of both the torch’s center position and height position, achieving synchronized torch pose detection with the arc position.

### 2.2. System Principle

To avoid the influence of arc motion and camera shake on image quality during detection and to improve the stability and adaptability of the visual sensing detection system, a synchronized image acquisition method with the arc position was established. When the arc swings to the left or right sidewall of the welding groove, the welding controller synchronously outputs the arc position signal *P*_arc_, which is used to trigger the camera to capture a frame of the welding image during the arc’s stay time at the left or right sidewall. This method ensures that the image is acquired when the arc is relatively stable, thereby improving image quality and detection accuracy.

Figure 2 illustrates the trigger signals for the arc position. When the arc motion signal is at a high level, the stepper motor is in motion; when it is at a low level, the stepper motor is in a stop state. Each time the arc swings to the left or right sidewall of the groove, the welding controller sends the arc positional signal *P*_arc_ to the camera, which then captures a frame of the welding image with the arc at the groove sidewall. These images are transmitted to the computer image processing system through the image acquisition card for the subsequent detection of the torch pose.

### 2.3. Welding Image Processing

Figure 3 illustrates the processing results of the welding image. During DC welding, the arc swing frequency is 2.5 Hz, the sidewall stay time is 100 ms, and the process gap is 2 mm. A dimming system, composed of a narrowband filter with a central wavelength of 970 ± 20 nm and a 10% neutral density filter, is coaxially connected to the camera. The experimental conditions for camera shooting and DC welding are listed in Table 1 and Table 2.

Adjacent frames of the welding images are captured when the arc swings to the left and right sidewalls of the groove, as shown in Figure 3(a_1_,a_2_). Utilizing the established passive visual sensing system, without the need for additional light sources, the captured welding images clearly display the arc, the left and right edges of the groove, the wire position, molten pool characteristics, and slag. This clarity provides excellent conditions for extracting welding image features.

To calculate the torch pose, it is crucial to extract the arc contour through image processing methods. First, the welding image is subjected to global threshold segmentation using a threshold value of 240, obtained through grayscale histogram analysis. The resulting threshold images are shown in Figure 3(b_1_,b_2_). Next, a morphological anti-interference extraction method with erosion and dilation operations is performed on these threshold images using a 3 × 3 rectangular structuring element to eliminate the influence of arc light reflection interference, thereby enhancing the accuracy of arc contour extraction. The results are displayed in Figure 3(c_1_,c_2_,d_1_,d_2_). Subsequently, the arc contour is extracted using the Canny edge detector in turn, as shown in Figure 3(e_1_,e_2_). Finally, an ellipse model-based arc contour fitting method is applied, with the fitted arc contour lines indicated by the red lines in Figure 3(f_1_,f_2_). Row and column scans are performed on the arc contour fitting lines to extract the coordinates of the arc’s highest point and center point. Coordinates in pixels (same below) of the arc’s highest points are (396, 179) and (161, 191), respectively, while the center coordinates of the arc contour are (400, 129) and (177, 132). The acquisition of features such as the arc’s highest point and the arc center coordinates provides a solid technical foundation for the efficient detection of the torch pose.

## 3. Detection Method for Torch Pose

Based on the extracted arc contour features, a torch pose detection (TPD) method is proposed to accurately detect the torch center position and torch height deviation. The TPD method includes obtaining the arc contour fitting line, detecting the arc center position, and determining the arc height position.

Figure 4 demonstrates the principle diagram of the TPD method. The infrared camera captures a frame of the welding image when the arc stays at the left sidewall of the groove. The height and width of the welding image are *h*_1_ and *w*_1_, respectively. The welding image is located in the *x_i_*-*O_i_*-*y_i_* coordinate system, with the *x_i_*-axis representing the width direction and the *y_i_*-axis representing the height direction. The welding image contains the left and right edges of the groove, the lead and tail of the pool, the wire, and the arc. The schematic welding image in the current frame is displayed, with the arc at the groove’s right sidewall in the adjacent previous frame. The wire and arc from the previous frame are shown with dashed lines in Figure 4.

The arc contour fitting line is extracted through welding image processing, and an ellipse model is used to obtain the arc contour fitting line. A row scan is performed on the arc contour fitting line to obtain the upper or lower limit point coordinates along the horizontal sequence. The upper limit point *A_ci_* is the highest point of the arc, with coordinates (*x_ci_*, *y_ci_*), and the highest point of the arc in the previous frame welding image is *A_c_*_(*i*−1)_, with coordinates (*x_c_*_(*i*−1)_, *y_c_*_(*i*−1)_). Then, a column scan is performed on the arc contour fitting line to search for the left and right limit point coordinates along the vertical sequence. The left and right limit points are connected, as well as the upper and lower limit points. The intersection of these two connecting lines is the arc contour center point *A_ai_*, with coordinates (*x_ai_*, *y_ai_*). The arc contour center point in the previous frame welding image is *A_a_*_(*i*−1)_, with coordinates (*x_a_*_(*i*−1)_, *y_a_*_(*i*−1)_).

By calculating the average value of the horizontal coordinates *x_ai_* and *x_a_*_(*i*−1)_ from the current frame arc contour center point *A_ai_* and the previous frame arc contour center point *A_a_*_(*i*−1)_, the torch center position sample value Δ*x_Ti_* is obtained, as shown in expression (1). By calculating the difference between the vertical coordinate *y_ci_* and *y_c_*_(*i*−1)_ from the current frame arc highest point *A_ci_* and the previous frame arc highest point *A_c_*_(*i*−1)_, the torch height deviation value Δ*y_vi_* is obtained, as shown in expression (2).
Δ*x_Ti_* = (*x_ai_* + *x_a_*_(*i*−1)_)/2(1)
Δ*y_vi_* = *y_ci_* − *y_c_*_(*i*−1)_(2)

Additionally, due to the influence of arc parameters, the highest point of the arc fluctuates significantly with random variations. To improve the detection accuracy of Δ*y_vi_*, a mean filtering method is applied to process multiple consecutive samples of *y_ci_* before calculating Δ*y_vi_*. After removing interference data, the detection values for the torch height deviation Δ*y_vi_* are obtained. When Δ*y_vi_* < 0, it indicates that the torch height decreases. When Δ*y_vi_* = 0, it indicates that the torch height remains unchanged. And, when Δ*y_vi_* > 0, it indicates that the torch height increases. Meanwhile, a mean filtering method is also applied to process multiple consecutive samples of Δ*x_Ti_* to obtain the detection value for the torch center Δ*x’_Ti_*.

## 4. Experimental Results

### 4.1. Experimental Methods and Conditions

To verify the adaptability and effectiveness of the proposed algorithm, torch pose detection experiments were conducted under conditions of both constant and variable groove width in swing arc narrow gap welding. Constant groove width welding is defined as the groove width remaining unchanged during the welding process. In contrast, variable groove width welding is defined as the groove width gradually changing during the welding process.

The welding testpieces were made of low-carbon steel and processed to obtain I-shaped grooves, as shown in Figure 5. *G_b_* was the initial width of the groove, *G_e_* was the final width of the groove, and *L*_4_ was the height of the testpiece. The groove depth *L*_1_ = *L*_4_/2. In the variable groove welding experiment, the groove width of the testpiece uniformly increased from *G_b_* = 12 mm to *G_e_* = 16 mm. The testpiece dimensions (*L*_2_ × *L*_3_ × *L*_4_) were 180 mm × 60 mm × 30 mm, with *L*_1_ = 15 mm. The swing angle was determined based on the value of *G_b_*, which was 68°. In the constant groove welding test, the groove width of the testpiece was fixed at *G_b_* = *G_e_* = 13.8 mm, and the swing angle *α* = 78°. The testpiece dimensions (*L*_2_ × *L*_3_ × *L*_4_) were 70 mm × 60 mm × 30 mm, with *L*_1_ = 15 mm.

During the experiment, the camera captured welding images at a resolution of 544 × 544 pixels to detail the arc and weld seam. The arc swing frequency was 2.5 Hz, the sidewall stay time was 100 ms, the bending angle of the conductive rod was 8°, the torch standoff height was 20 mm, the brand name of the wire was ER49-1, and the process gap was 2 mm. Except for a slightly increased camera depression angle, other conditions were the same as those shown in Table 1 and Table 2.

### 4.2. Detected Results of Torch Pose in Width-Varying Groove Welding

In the variable groove welding experimental, the distance of the arc to the sidewalls of the groove was uneven due to the varying groove width. The arc was ignited in the groove of the testpiece. Once the arc stabilized, welding images were captured starting from a position approximately 20 mm from the beginning of the groove. The arc extinguished approximately 20 mm from the end of the groove, and image capture was then stopped. A total of 215 welding images were captured, with coordinates located in the *x_i_*-*O_i_*-*y_i_* coordinate system. 

Figure 6 gives the fitting results of arc contours for the width-varying groove. Two groups of adjacent welding images were located at approximately 54 mm and 156 mm along the groove length, as shown in Figure 6(a_1_–a_4_). When the swing angle of the arc in the groove remained constant, the distance from the arc to the groove sidewall gradually increased as the groove width increased. According to the image processing method described in Figure 3, the threshold images were extracted using a threshold of 240. After performing morphological anti-interference extraction on the threshold images, the Canny operator was used to extract the arc contour. 

The arc contour fitting lines were obtained using the fitting method and are shown as red lines in Figure 6(b_1_–b_4_). Row and column scans were performed on the arc contour fitting lines to extract the coordinates of the arc’s highest point *A_ci_*, which were (182, 206), (340, 195), (176, 202), and (334, 189), respectively. The arc contour center point (*A_ai_*) coordinates were (202, 157), (341, 155), (197, 159), and (339, 149), respectively. These results indicated that the image processing and fitting methods could accurately extract the arc contour and its key point coordinates, providing reliable data for torch pose detection.

To verify the accuracy and adaptability of the TPD method in detecting the torch center position, 215 arc contours and their fitting lines were extracted from the collected 215 welding images. By scanning the arc contour fitting lines, the horizontal coordinates *x_ai_* of the arc contour center points in each welding image frame were calculated, as shown by the green hollow circles in Figure 7. By calculating the average horizontal coordinates of the arc contour center points in two adjacent welding images, the torch center position sample value Δ*x_Ti_* was obtained, as shown by the blue squares in Figure 7. By applying mean filtering to the consecutive samples of five arc center positions, the detected torch center position values Δ*x’_Ti_* were obtained using the TPD method, as shown by the red solid squares in Figure 7. At this point, the detection range of the torch center position was 14.152 to 14.463 mm, with a range of 0.311 mm. The standard deviation of the detected values was 0.056. This indicated that the welding process was stable and the dynamic fluctuation range of the arc center position was small.

To further verify the detection accuracy of the TPD method, it was compared with the conventional torch center detection (CTCD) method [31]. The specific steps of the CTCD method involve setting the wire ROI window after extracting the arc contour, extracting the fitting lines of the adjacent two wire centers through image processing, and fitting them to a fixed position in the image to calculate the average wire center line of the corresponding column positions. In the experiment, the torch center position sample value was obtained by fitting to column 350, and the detected torch center position value Δ*x’_Ti_* obtained by applying the mean filtering algorithm is shown by the black hollow triangles in Figure 7. The detection error range of the CTCD method was 14.422 to 14.850 mm, with a range of 0.428 mm. The standard deviation of the detected values was 0.060, as listed in Table 3.

The comparative analysis results indicated that the detection accuracy of the torch center position using the CTCD method was relatively low. This is because, at the beginning of the welding process, the pool did not fully cover the entire weld seam. Consequently, the wire ROI image was affected by the low proportion of the pool background, resulting in reduced accuracy in the extraction and detection of the wire contour. Therefore, in the first five frames of the welding images at the beginning of the welding process, the CTCD method could not extract a complete wire image due to the influence of the welding image background, rendering the torch position detection ineffective. Additionally, a non-detection zone existed at the beginning of the welding, approximately 10 mm from the groove length. The insufficient pool background caused significant fluctuations in the detection accuracy of the torch center position. Thus, the CTCD method was more sensitive to the fitting position of the wire ROI window and the straightness of the wire during the torch center position detection, resulting in higher values of Δ*x’_Ti_* compared to the TPD method. This also led to lower detection accuracy at the beginning of welding.

Compared to the CTCD method, the TPD method enhanced the existing algorithm by reducing dependence on the molten pool background in the welding images. It avoided the need to calculate the wire center after extracting the arc contour to obtain the torch center position, thus reducing the impact of wire straightness on the detection accuracy of the torch center position. By using the TPD method, the accuracy and adaptability of the algorithm in detecting the torch center position were verified.

Moreover, the TPD method could detect not only the torch center position but also the torch height position. Figure 8 displays the detected results of torch height deviation for the width-varying groove. Based on the 215 welding images obtained, the arc contour fitting lines were extracted and the vertical coordinate sample values *y_ci_* of the arc’s highest points were obtained, as shown by the black solid squares in Figure 8a.

By applying mean filtering to the consecutive samples of five arc highest point vertical coordinates, the detected vertical coordinate values of the arc highest points were obtained, as shown by the red solid circles in Figure 8a. The fluctuation range of the mean-filtered *y_ci_* values decreased from 12.209 to 10.356 mm, resulting in a total range reduction of 1.853 mm. By calculating the difference between the mean-filtered values of *y_ci_* in two adjacent welding images, the detected torch height deviation Δ*y_vi_* ranged from −0.964 to +0.750 mm, as shown in Figure 8b. Meanwhile, the accumulation of Δ*y_vi_* is shown by the blue hollow circles in Figure 8a. Thus, as the groove width increased, the arc height gradually decreased, leading to an increasing cumulative deviation in torch height.

Therefore, when the groove width gradually increased from 12 to 16 mm, if the wire feeding speed, welding speed, and arc swing angle were not adjusted accordingly, the welding deposition amount decreased, causing the highest point of the arc to gradually move downward. This means it is necessary to monitor the torch height position variation based on the detected vertical coordinate values of the arc’s highest points. This allows for adjusting the torch height position and to achieve an adaptive adjustment of the torch pose in the narrow gap welding process.

### 4.3. Detected Results of Torch Pose for Constant Groove Welding

To verify the adaptability and effectiveness of the proposed algorithm, torch pose detection experiments were conducted under the condition of constant groove width in swing arc narrow gap welding. A total of 90 frames of constant groove welding images were collected. Figure 9 presents the welding images from experiments conducted with a constant-width groove.

Figure 9a gives the first frame of the image, where there is no welding pool as the background, increasing the difficulty of extracting the wire with the existing CTCD algorithm. However, the clear arc shape in the image does not affect the proposed TPD algorithm for obtaining the torch position based on arc features. During the experiment, the arc was initiated within the groove approximately 2 mm from the beginning of the groove. Once the arc stabilized, welding images were captured. The arc was extinguished approximately 7 mm from the end of the groove, terminating the collection of welding images. Figure 9b,c show the eighth and ninth welding images, located approximately 8 mm along the groove length. Figure 9d,e show the 73rd and 74th welding images, located approximately 52 mm along the groove length.

Figure 10 shows the detection results of the torch center position. During the welding process, starting from the first frame of the welding image, a total of 90 welding images were collected along the groove length, and the arc contours were extracted 90 times, accordingly. This resulted in 45 horizontal coordinates for the left center point of the arc contour and 45 horizontal coordinates for the right center point of the arc contour, as shown by the green hollow circles in Figure 10.

By calculating the average horizontal coordinates of the arc contour center positions in two adjacent welding image frames, 89 torch center position sample values Δ*x_Ti_* were obtained, as shown by the blue squares in Figure 10. By applying mean filtering to every five adjacent torch center position sample values, the detected torch center position values Δ*x’_Ti_* were obtained using the TPD method, as shown by the red solid squares in Figure 10. The constant groove welding torch center position detection results showed that the values of Δ*x’_Ti_* had a fluctuation range of 14.083 to14.399 mm, with a total range of 0.316 mm. In both the constant and variable groove welding experiments, the detected error range for the torch center position was within 0.32 mm.

Figure 11 shows the torch height deviation in the constant groove width test. During the welding process, 90 vertical coordinate sample values *y_ci_* of the arc highest points were extracted, as shown by the black solid squares in Figure 11a. By applying mean filtering to every five adjacent vertical coordinate sample values, the detected vertical coordinate values of the arc‘s highest points were obtained, as shown by the red solid circles in Figure 11a. By calculating the difference between the filtered vertical coordinate detection values of the arc’s highest points in two adjacent welding images, 89 torch height deviation values Δ*y_vi_* were obtained, as shown in Figure 11b. Similarly, the accumulation values of Δ*y_vi_* are shown by the blue hollow circles in Figure 11a.

Compared to Figure 8b, the arc height deviation shown in Figure 11b has significantly stabilized, indicating that the arc state was more stable during the welding process with a constant groove width, because the arc swing angle and other parameters were not adjusted in real-time when welding with variable groove widths. This also demonstrates the effectiveness of the TPD method for monitoring changes in the arc height position. The experimental results showed that the torch height deviation detected by the TPD method ranged from −0.225 to +0.236 mm. These results demonstrate that the proposed TPD algorithm can simultaneously detect the changes in both the torch center and torch height positions, providing a theoretical basis for achieving the synchronized detection of the torch’s horizontal and vertical positions.

## 5. Conclusions

This study aims to enhance welding quality through real-time torch pose detection, enabling precise adjustments based on welding parameters. A passive vision-based torch pose detection (TPD) method is proposed, which improves upon existing algorithms by eliminating the need to calculate the wire center to obtain the torch center position, thus reducing the impact of wire straightness on detection accuracy. Additionally, the method decreases reliance on image background and grayscale differences, achieving synchronous detection of the torch’s horizontal and height deviations. The main conclusions are as follows:An elliptical model-based arc contour fitting method was proposed, which improves the detection accuracy of arc features by obtaining a fitted arc contour line.A method for detecting the torch center position was developed by scanning the limit points of the arc contour fitting line.A torch height position detection method was proposed by scanning the highest points of the arc contour fitting line.

## Figures and Tables

**Figure 1 sensors-24-04996-f001:**
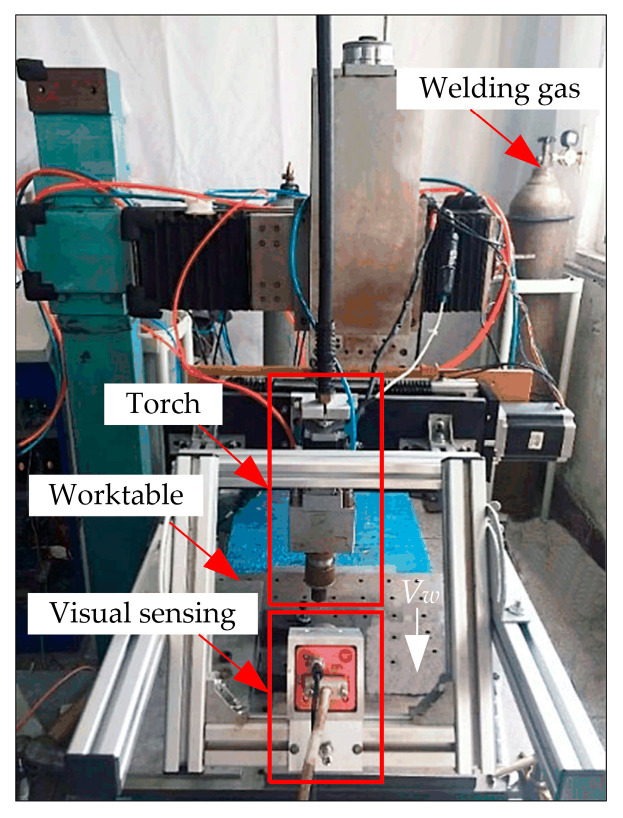
Passive visual sensing detection system of torch position.

**Figure 2 sensors-24-04996-f002:**
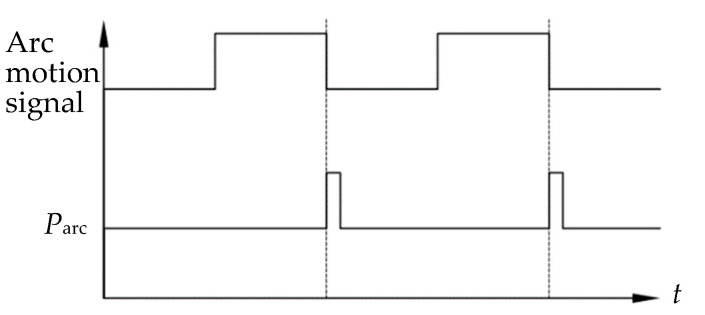
Trigger signals of arc position.

**Figure 3 sensors-24-04996-f003:**
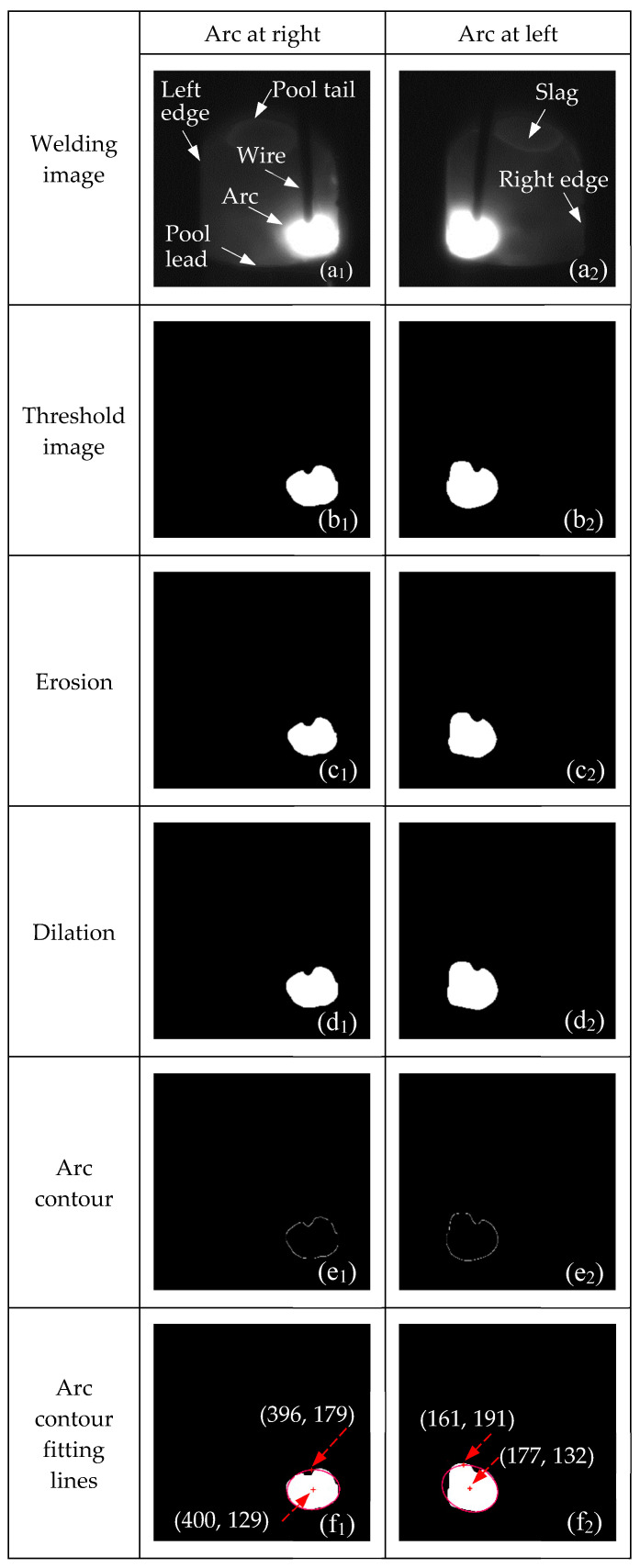
Processing results of welding image.

**Figure 4 sensors-24-04996-f004:**
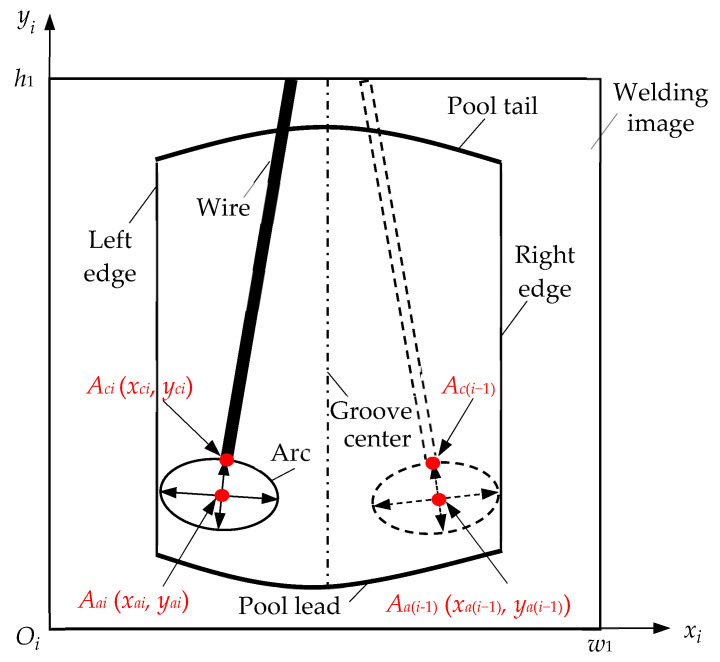
Principle of torch position detection.

**Figure 5 sensors-24-04996-f005:**
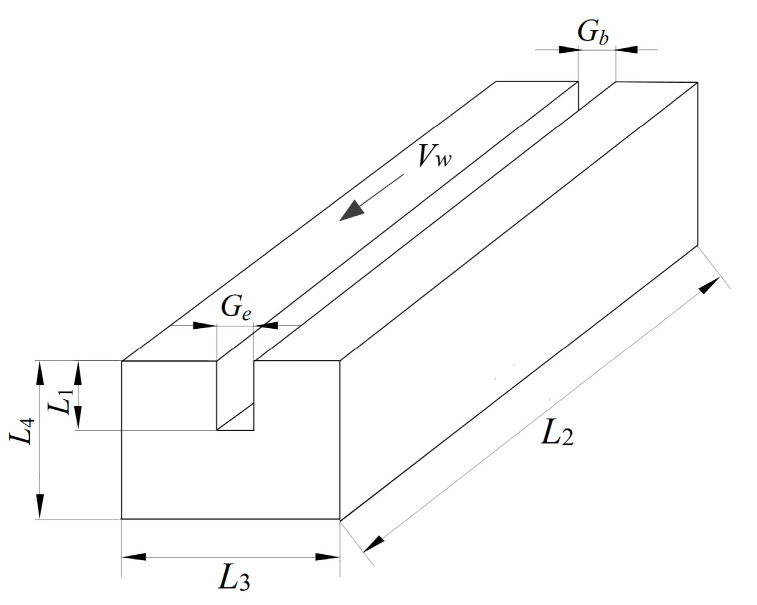
Schematic diagram of variable groove.

**Figure 6 sensors-24-04996-f006:**
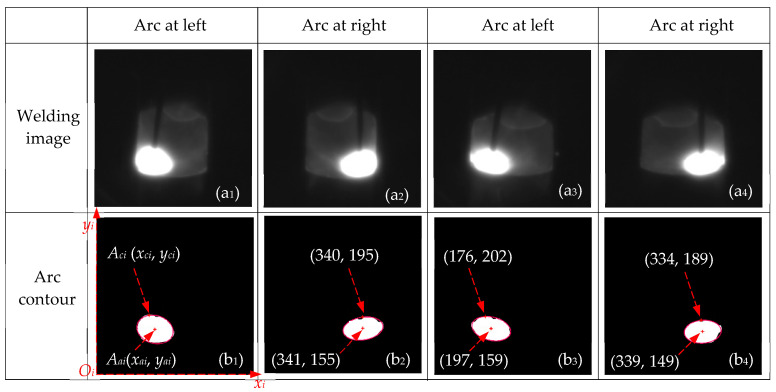
Fitting results of arc contours for the width-varying groove.

**Figure 7 sensors-24-04996-f007:**
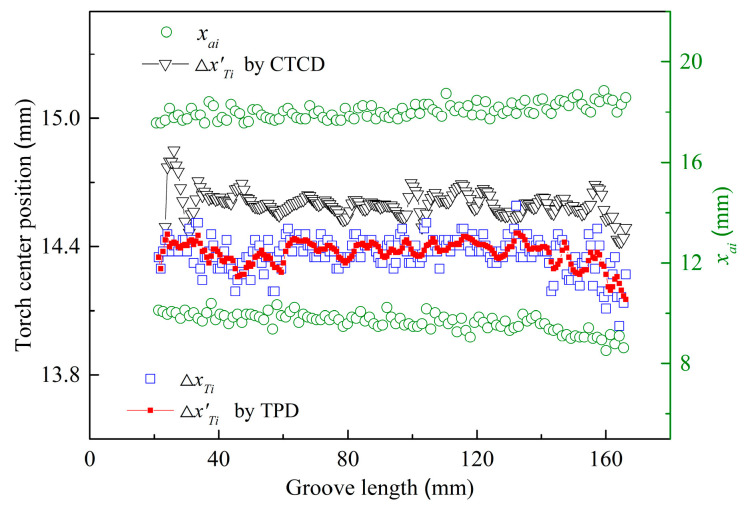
Detected results of torch center position for width-varying groove.

**Figure 8 sensors-24-04996-f008:**
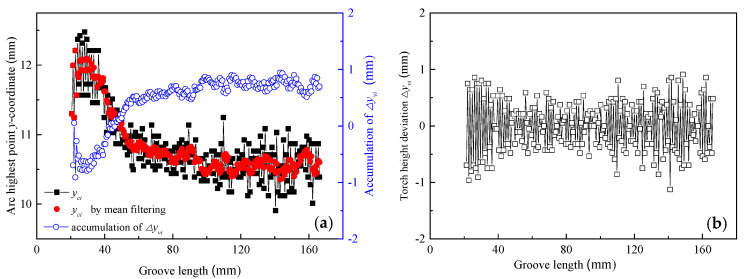
Detected results of torch height deviation for the width-varying groove: (**a**) *y*-coordinate of the arc highest point and accumulation of Δ*y_vi_*; (**b**) torch height deviation.

**Figure 9 sensors-24-04996-f009:**
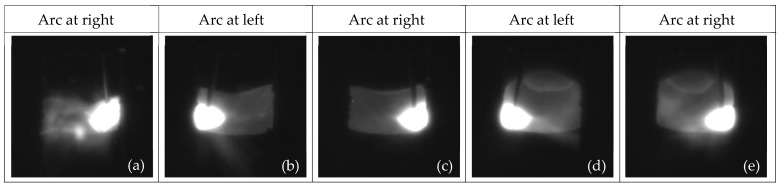
Welding images for constant-width groove.

**Figure 10 sensors-24-04996-f010:**
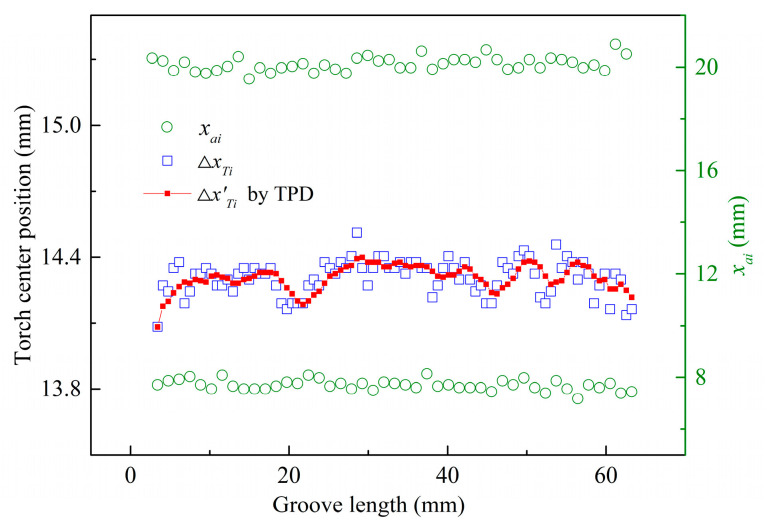
Detected results of torch center position for constant-width groove.

**Figure 11 sensors-24-04996-f011:**
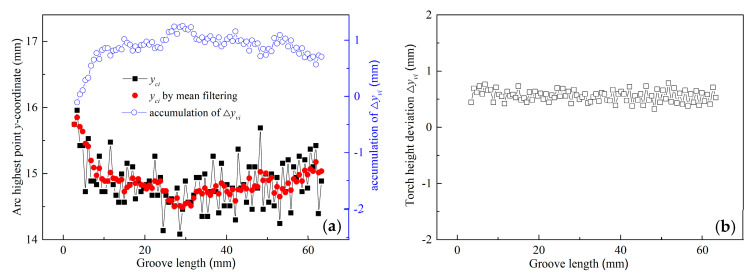
Detected results of torch height deviation for the constant-width groove: (**a**) *y*-coordinate of the arc highest point and accumulation of Δ*y_vi_*; (**b**) torch height deviation.

**Table 1 sensors-24-04996-t001:** Parameters of camera shooting.

Parameter Name	Value
Central wavelength of narrowband filter (nm)	970 ± 20
Neutral density filter (%)	10
Aperture	f/11
Exposure time (ms)	3
Shooting depression angle (°)	25
Global image size (pixel)	544 × 544

**Table 2 sensors-24-04996-t002:** DC welding conditions.

Parameter Name	Value
Arc current (A)	290
Arc voltage (V)	28
Welding speed *V_w_* (mm min^−1^)	204
Solid wire diameter (mm)	1.2
Torch standoff height (mm)	20
Shielding gas/flowrate (L min^−1^)	Ar − 20%CO_2_/25
Arc swing frequency (Hz)	2.5
Arc at-sidewall staying time (ms)	100
Conductive-rod bending angle (°)	8

**Table 3 sensors-24-04996-t003:** Comparison of detected results.

Method	Range (mm)	Standard Deviation	Non-Detection Zone
TPD	0.311	0.056	~10 mm
CTCD	0.428	0.060	0 mm

## Data Availability

Data are contained within the article.

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
