# Peer review of "Passive Vision Detection of Torch Pose in Swing Arc Narrow Gap Welding"

_sensors, 2024, doi:10.3390/s24154996_

Round 1
Reviewer 1 Report
Comments and Suggestions for Authors
The manuscript is proposed a torch pose detection method to realize the simultaneous detection of torch center and height positions for swing arc narrow gap welding.
1- The abstract needs to be summarized again, think about whether it's better to use the past tense in the abstract.
2- Ensure that all tables are clearly. For example, in Table 1 of Section 2.3, the data for "Central wavelength of narrow filter" should be consistent with the content of the paper.
3- Improve the logical structure in section 4.4 of the article. The contents in lines 383-386, which pertain to welding conditions, should be moved to the first paragraph of section 4.4.
4- The format of the references needs to be carefully revised to ensure compliance with the journal requirements. Additionally, check for duplicate references and remove any redundancies.
5-The Conclusion should primarily highlight the key findings of the study. Currently, it appears overly lengthy and includes several repetitions of previously discussed content.
Comments on the Quality of English LanguageThe English expression of the entire article is relatively easy to understand. Please double check and review the tense of the entire text.
Author Response
Original paper number: Sensors-3133392
Title: Passive vision detection of torch pose in swing arc narrow gap welding
Responses to reviewers
Reviewer #1: The manuscript is proposed a torch pose detection method to realize the simultaneous detection of torch center and height positions for swing arc narrow gap welding.
Comment 1: The abstract needs to be summarized again, think about whether it's better to use the past tense in the abstract.
Response1: Thanks to the reviewer for the suggestion. We have summarized the abstract again and considered your suggestion to use the past tense. The necessary changes have been made accordingly. We appreciate your detailed review and helpful comments.
Comment 2: Ensure that all tables are clearly. For example, in Table 1 of Section 2.3, the data for "Central wavelength of narrow filter" should be consistent with the content of the paper.
Response 2: Thank you for your valuable suggestions. We have ensured that all tables are clearly presented. For example, in Table 1 in line one, the data for "Central wavelength of narrow filter" has been updated to be consistent with the content of the paper with 970±20. The 0.1s rewrite 100ms in Table 2. Thank you for bringing this to our attention.
Comment 3: Improve the logical structure in section 4.4 of the article. The contents in lines 383-386, which pertain to welding conditions, should be moved to the first paragraph of section 4.4.
Response 3: We have improved the logical structure in section 4.4 of the article. Specifically, the contents of welding conditions, have been moved to the first paragraph of section 4.4. We appreciate your detailed review and helpful suggestions. Additionally, we have revised the manuscript to merge sections 4.2, 4.3, and 4.4 into two sections, 4.2 and 4.3, for better coherence and clarity.
Comment 4: The format of the references needs to be carefully revised to ensure compliance with the journal requirements. Additionally, check for duplicate references and remove any redundancies.
Response 4: We have carefully revised the format of the references to ensure compliance with the journal requirements. Additionally, we have checked for duplicate references and removed any redundancies. We appreciate your detailed review and helpful suggestions.
Comment 5: The Conclusion should primarily highlight the key findings of the study. Currently, it appears overly lengthy and includes several repetitions of previously discussed content.
Response 5: Thank you for your valuable feedback. We have revised the Conclusion to primarily highlight the key findings of the study. The section has been condensed to eliminate repetitions and ensure clarity. We appreciate your detailed review and constructive suggestions.

Reviewer 2 Report
Comments and Suggestions for Authors
This paper proposed a method for detecting the torch pose in narrow gap welding using infrared visual sensing. I have a few questions need the authors to explain or present more clearly.
Point 1:Authors are advised to optimize the abstract and further clarify the highlights of the study.
Point 2: Please add units after the number to make it clearer for the reader. For example, in section 2.2, “The coordinates of the highest points of the arc are (396,179) and (161,191), respectively,” should indicate whether these numbers are “pixel” of “mm”.
Point 3: In section 4.2, titled "Image processing", should be merged with section 4.3, and both should be combined under section 4.2 for better clarity and structure.
Point 4: The reference section should be expanded to include over 30-50 published papers, emphasizing the most recent research advancements from the past 5 years.
Author Response
Comment 1: Authors are advised to optimize the abstract and further clarify the highlights of the study.
Response 1: Thank you for your insightful feedback. We have optimized the abstract and further clarified the highlights of the study to ensure it effectively summarizes our key findings. We appreciate your detailed review and helpful suggestions.
Comment 2: Please add units after the number to make it clearer for the reader. For example, in section 2.2, “The coordinates of the highest points of the arc are (396,179) and (161,191), respectively,” should indicate whether these numbers are “pixel” of “mm”.
Response 2: We have added units after the numbers to make them clearer for the reader. For example, in section 2.2, we have specified that “The coordinates of the highest points of the arc are (396,179) and (161,191), respectively,” are in pixels. Example in lines 174~175.
Comment 3: In section 4.2, titled "Image processing", should be merged with section 4.3, and both should be combined under section 4.2 for better clarity and structure.
Response 3: We have merged section 4.2, titled "Image processing," with section 4.3, and combined both under section 4.2 for better clarity and structure. Example in line 244.
Comment 4: The reference section should be expanded to include over 30-50 published papers, emphasizing the most recent research advancements from the past 5 years.
Response 4: Thanks to the reviewer for the suggestion. We have expanded the reference section to include over 30-50 published papers, emphasizing the most recent research advancements from the past 5 years. We value your thorough review and insightful recommendations.

Reviewer 3 Report
Comments and Suggestions for Authors
An original article is presented, which is aimed at passive vision detection of torch pose in swing arc narrow gap welding. This line of research is very important because the quality of the weld depends on the accuracy of the torch positioning. Moreover, this direction is important not only for welding, but also for surfacing operations using the electric arc surfacing method and electric spark alloying. In the article, the adequacy of the developed methodology is proven by experimental results. The article contains new technical and scientific results, which makes it interesting for the reader. For a better understanding of the article, I suggest that the authors pay attention to the comments made below.
As already mentioned above, adjusting the position of the torch in height during welding and electric arc surfacing is an important area. I recommend that this importance be reflected in lines 42-43 in the form of additional references to relevant works: https://doi.org/10.1016/j.surfcoat.2021.127952, https://doi.org/10.3103/S1068375511040107
At the end of the introduction, add the purpose of the work.
Considering that during welding there is scattering of the electrode and substrate material in the form of drops and sparks, especially at elevated conditions, it is necessary to indicate whether the chamber was in any way protected from this impact. This is important because once the chamber is clogged, it will not function.
Table 2 shows the welding modes, have other modes been studied? If not, then the article can provide the authors’ reasoning about their impact on the accuracy of the proposed image processing.
In the experimental work carried out, low carbon steel was used as a substrate. The question arises: will the developed methodology work adequately for HSS? I propose to add this reasoning to the article and conclusion.
Author Response
Comment 1: As already mentioned above, adjusting the position of the torch in height during welding and electric arc surfacing is an important area. I recommend that this importance be reflected in lines 42-43 in the form of additional references to relevant works:
https://doi.org/10.1016/j.surfcoat.2021.127952, https://doi.org/10.3103/S1068375511040107
Response 1: Thank you for your valuable feedback. We have addressed the importance of adjusting the position of the torch during welding and electric arc surfacing in lines 42-43 (Current version in line 45) by adding references to the relevant works you suggested:
https://doi.org/10.1016/j.surfcoat.2021.127952, and https://doi.org/10.3103/S1068375511040107
Comment 2: At the end of the introduction, add the purpose of the work.
Response 2: We have added the purpose of the work at the end of the introduction to clearly outline the objectives of our study.
Comment 3: Considering that during welding there is scattering of the electrode and substrate material in the form of drops and sparks, especially at elevated conditions, it is necessary to indicate whether the chamber was in any way protected from this impact. This is important because once the chamber is clogged, it will not function.
Response 3: We appreciate the reviewer’s insightful comment regarding the potential impact of scattering electrode and substrate material during welding, especially at elevated conditions. It is indeed crucial to ensure that the welding chamber remains protected from such impacts to maintain its functionality.
To address this concern, we have implemented a filtration system, protective shields, and a ventilation system as protective measures in our welding setup. In conclusion, these protective measures have proven to be effective in safeguarding the chamber from the impact of scattering electrode and substrate material, ensuring the chamber's continued functionality and reliability under elevated welding conditions.
Comment 4: Table 2 shows the welding modes, have other modes been studied? If not, then the article can provide the authors’ reasoning about their impact on the accuracy of the proposed image processing.
Response 4: Thank you for your insightful suggestion. Figure 2 illustrates an image acquisition mode to capture welding images when the arc reaches and stays at the left or right sidewall of the groove, reducing the impact of arc light on the opposite edge of the groove. Additionally, the article can still accurately extract the arc contour and detect the torch pose regardless of the arc's position within the sidewall of groove, as demonstrated by its adaptability when the arc is positioned at the center of the groove. Of course, the arc contour and torch pose can be detected regardless of the arc's position.
Comment 5: In the experimental work carried out, low carbon steel was used as a substrate. The question arises: will the developed methodology work adequately for HSS? I propose to add this reasoning to the article and conclusion.
Response 5: Thanks for your valuable feedback. The proposed method is for detecting torch pose, and the arc shape is not significantly affected by the type of steel, whether it is low carbon steel or HSS. Therefore, the proposed method can identify the arc contour and obtain the torch pose regardless of the steel type.
